

# Disruption of *de novo* purine biosynthesis in *Pseudomonas fluorescens* Pf0-1 leads to reduced biofilm formation and a reduction in cell size of surface-attached but not planktonic cells

Shiro Yoshioka[1] and Peter D. Newell[2]

[1] Department of Life and Coordination-Complex Molecular Science, Institute for Molecular Science, National Institutes of Natural Sciences, Okazaki, Aichi, Japan

[2] Department of Biological Sciences, Oswego State University of New York, Oswego, NY, USA

## ABSTRACT

*Pseudomonas fluorescens* Pf0-1 is one of the model organisms for biofilm research. Our previous transposon mutagenesis study suggested a requirement for the *de novo* purine nucleotide biosynthesis pathway for biofilm formation by this organism. This study was performed to verify that observation and investigate the basis for the defects in biofilm formation shown by purine biosynthesis mutants. Constructing deletion mutations in 8 genes in this pathway, we found that they all showed reductions in biofilm formation that could be partly or completely restored by nucleotide supplementation or genetic complementation. We demonstrated that, despite a reduction in biofilm formation, more viable mutant cells were recovered from the surface-attached population than from the planktonic phase under conditions of purine deprivation. Analyses using scanning electron microscopy revealed that the surface-attached mutant cells were $25 \sim 30\%$ shorter in length than WT, which partly explains the reduced biomass in the mutant biofilms. The laser diffraction particle analyses confirmed this finding, and further indicated that the WT biofilm cells were smaller than their planktonic counterparts. The defects in biofilm formation and reductions in cell size shown by the mutants were fully recovered upon adenine or hypoxanthine supplementation, indicating that the purine shortages caused reductions in cell size. Our results are consistent with surface attachment serving as a survival strategy during nutrient deprivation, and indicate that changes in the cell size may be a natural response of *P. fluorescens* to growth on a surface. Finally, cell sizes in WT biofilms became slightly smaller in the presence of exogenous adenine than in its absence. Our findings suggest that purine nucleotides or related metabolites may influence the regulation of cell size in this bacterium.

Subjects Biochemistry, Microbiology
Keywords *De novo* purine nucleotide biosynthesis, Cell size, Biofilm, Nutrient deprivation

## INTRODUCTION

ATP and GTP are the purine nucleotide triphosphates that are essential to drive many cellular processes in all living organisms. ADP and GDP are utilized as DNA precursors

Corresponding author
Shiro Yoshioka, yoshioka@ims.ac.jp

after being converted to the deoxy forms by ribonucleotide reductase (*Neuhard & Nygaar, 1987*). AMP and GMP are the dephosphorylated forms of the above nucleotides and synthesized either in a *de novo* synthesis pathway or in a salvage pathway (*Neuhard & Nygaar, 1987*). In the *de novo* purine biosynthesis pathway, inosine monophosphate (IMP) is sequentially synthesized from 5-phosphoribosyl-$\alpha$-diphospahte (PRPP) in 11 enzymatic steps, where some reactions require ATP to proceed. AMP and GMP are synthesized separately from IMP as the common intermediate. Therefore, the biosynthesis of purine nucleotides has a high energy cost. For this reason, another biosynthesis pathway, salvage pathway, is designed to scavenge and recycle the purine bases arising from nucleic acid turnover; adenine, guanine, and hypoxanthine are converted into AMP, GMP and IMP, respectively, by phosphoribosyltransferases (*Neuhard & Nygaar, 1987*). The fact that most bacteria possess both *de novo* purine biosynthesis pathway and salvage pathway indicates vital role of this pathway in bacteria.

The importance of the *de novo* purine biosynthesis in bacterial growth has been repeatedly described in the literature. If one of the genes in *de novo* purine biosynthesis pathway is disrupted, the mutant becomes purine auxotroph. In other words, the mutant is not able to grow unless the exogenous purine bases such as adenine and hypoxanthine are supplied. Purine requiring mutants of some pathogenic bacteria have been found to be avirulent in murine models of infection, implying that the purine requiring mutants stop growing when exogenous purines are not available at the sites of infection, leading to attenuated infection (*Bacon, Burrows & Yates, 1951*; *Gerber, Hackett & Franklin, 1952*; *Straley & Harmon, 1984*; *Wang et al., 1996*; *Polissi et al., 1998*; *Pilatz et al., 2006*; *Samant et al., 2008*; *Jenkins et al., 2011*). Furthermore, recent research has highlighted the role of the purine nucleotide biosynthesis on biofilm formation and symbiosis with nematode, insect or plant roots (*Han et al., 2006*; *Ge et al., 2008*; *An & Grewal, 2011*; *Kim et al., 2014a*). In these studies, significant reductions in biofilm formation and defects in symbiotic ability were observed for the purine auxotrophic mutants, emphasizing important roles of the purine biosynthesis pathway in biofilm formation and symbiosis (*Han et al., 2006*; *An & Grewal, 2011*; *Ge et al., 2008*; *Kim et al., 2014a*).

The purine nucleotide derivative c-di-GMP is a central player in the regulation of biofilm formation. Generally, increase in cellular level of c-di-GMP facilitates biofilm formation. This compound is synthesized from two molecules of GTP by diguanylate cyclases (DGCs) possessing GGDEF domain (*Paul et al., 2004*; *Ryjenkov et al., 2005*), and degraded by phosphodiesterases (PDEs) containing either EAL or HD-GYP domain (*Christen et al., 2005*; *Schmidt, Ryjenkov & Gomelsky, 2005*; *Ryan et al., 2006*). As these enzymes typically contain regulatory domains, the synthesis and degradation of c-di-GMP is influenced by environmental factors.

While DGCs and PDEs have the primary role in controlling the c-di-GMP level, previous work suggested that nucleotide pools impact c-di-GMP levels (*Monds et al., 2010*; *Kim et al., 2014b*). The disruption of the *purT* gene in the *de novo* purine biosynthesis pathway decreased cellular concentration of c-di-GMP, leading to defect in the biofilm formation by *Burkholderia* species (*Kim et al., 2014b*). Increase in c-di-GMP level was observed for the *apaH* mutant of *Psudomonas fluorescens* Pf0-1, which is caused by promotion of the *de*

*novo* purine biosynthesis through increased level of di-adenosine tetraphosphate (Ap4A) (*Monds et al., 2010*). As for the purine auxotrophic mutants of *P. fluorescens*, biofilm formation could be impaired because the cellular concentrations of c-di-GMP may be decreased due to a reduced pool of GTP.

*P. fluorescens* Pf0-1 is a soil bacterium that promotes plant growth by forming biofilms on roots (*Haas & Defago, 2005*). This bacterium forms a biofilm using LapA, a large cell–surface adhesion protein with a molecular weight of ∼520 kDa (*Hinsa et al., 2003*). The secretion and localization of LapA is regulated by LapD that binds c-di-GMP (*Monds et al., 2007*; *Newell, Monds & O'Toole, 2009*). To identify the DGC genes involved in biofilm formation by this bacterium, we previously performed transposon mutagenesis and found that more than 50 genes are involved in the biofilm formation (*Newell et al., 2011*). Among the mutants impaired in biofilm formation, the transposon insertions occurred in the *purH*, *purL*, *purM*, *purF*, and *purK* genes in *de novo* purine nucleotide biosynthesis pathway (Table S3 in *Newell et al., 2011*). We therefore hypothesized that the *de novo* purine biosynthesis pathway is essential for the biofilm formation by this bacterium, as reported for other bacteria.

In this study, we sought to determine the basis for the biofilm formation defects observed in these mutants. To verify the requirement of the purine biosynthesis genes in biofilm formation, we constructed clean deletion mutants and performed functional complementation with exogenous genes and purine bases. Using electron scanning microscopy and a laser diffraction particle analyzer, we demonstrate that the surface-attached cells have smaller cell size compared to the planktonic cells and that the biofilm cells for the purine-depleted mutants became smaller than the WT cells. These data suggest purine auxotrophs of *P. fluorescens* are capable of surface attachment, but produce less biofilm biomass due to the impact of purine deprivation on growth and cell size.

## MATERIALS & METHODS

### Strains and media

Strain SMC4798 was used as the wild-type (WT) *P. fluorescens* Pf0-1, which expresses fully functional three-hemagglutinin (HA)-tagged LapA (*Monds et al., 2007*; *Newell, Monds & O'Toole, 2009*; *Newell et al., 2011*; *Boyd et al., 2014*). *Escherichia coli* S17-1 (λpir) was used for cloning and conjugation. *P. fluorescens* and *E. coli* were routinely cultured with LB medium in a test tube or on a solidified LB medium with 1.5% agar at 30 °C and 37 °C, respectively. *Saccharomyces cerevisiae* strain InvSc1 (Life Technologies, Carlsbad, CA, USA) was used to construct plasmids for clean deletion and complementation, as previously described (*Shanks et al., 2006*). Gentamycin (Gm) was used at 30 µg/ml for *P. fluorescens* and at 10 µg/ml for *E. coli*. Chloramphenicol (Cm) was used at 30 µg/ml. For biofilm assay, K10T-1 medium that has been used in this laboratory and consisted of 50 mM Tris–HCl (pH 7.4), 0.2% Bacto tryptone, 0.15% glycerol, 0.61 mM $MgSO_4$, and 1 mM $K_2HPO_4$, was used (*Monds et al., 2006*). Purine bases were added to K10T-1 to a final concentration of 0.2 mM. Arabinose was used to induce expression of the $P_{BAD}$ promoter from pMQ72 vector at a final concentration of 0.1% (wt/vol).
## Biofilm formation assay

Biofilm formation assays were performed using a polyvinyl chloride 96-well round-bottom microtiter plate (Corning 2797). Aliquots (1.5 µl) of liquid cultures grown overnight in LB medium was added to 100 µl of K10T-1 medium in the microtiter plate, and statically incubated at 30 °C for 6 h. After incubation, the liquid cultures were discarded, and the biofilm cells were stained with 0.1% (wt/vol) crystal violet (CV) in water. Twenty minutes later, the microtiter plates were rinsed with water three times, and air dried. Quantification of the biofilm cells was performed as previously described (*Monds et al., 2007*). In brief, 150 µl of 30% acetic acid (vol/vol) was added to the microtiter plate to solubilize the CV, and 125 µl of this solution was transferred into a flat-bottom microtiter plate. A microplate reader Vmax (Molecular Devices, Sunnyvale, CA, USA) was used to read the absorbance at 550 nm.

## Constructs for clean deletion and complementation

The pMQ30 and pMQ72 vectors were used for clean deletion and complementation, respectively. Clean deletion mutants were prepared as follows. A ∼1 kbp pair of PCR fragments was amplified from upstream and downstream of the target genes. The fragments were cloned into pMQ30 in parallel. Deletion constructs were transferred into *P. fluorescens* by conjugation, and transconjugates were selected on LB plates containing Gm and Cm. After confirming the single-crossover events by PCR, the strains were cultured overnight in the absence of any antibiotics and then spread on LB medium agar plate containing 5% (wt/vol) sucrose to facilitate the second crossover recombination. The removals of the target genes were verified by PCR, and all mutants resulted in the reduced biofilm in the biofilm formation assay. The pMQ72 vectors for complementation were designed to possess the ribosomal binding site (RBS) for LapD (*Newell, Monds & O'Toole, 2009*) and the N-terminal 6× histidine (6× His) tag. Success in construction was confirmed by DNA sequencing. The pMQ72 vectors were introduced into the *P. fluorescens* strains by electroporation.

## Measurement of numbers of planktonic and biofilm cells

Number of cells in the supernatants and biofilms were determined by a serial dilution method. After incubation for biofilm assay, the supernatants were removed, and the wells were washed twice with 100 µl of PBS buffer consisting of 10 mM $Na_2HPO_4$, 2 mM $KH_2PO_4$, 137 mM NaCl, and 2.7 mM KCl. An aliquot (100 µl) of the PBS buffer was added to the wells, and the biofilm cells were collected using a disposal cotton swab (*Shimada et al., 2012*). The cotton swab was then put into an Eppendorf tube, and the well was washed twice with 100 µl of the PBS buffer. The washed buffers were transferred into the Eppendorf tube each time. After adjusted to 1mL, the solution was vigorously vortexed and used for the serial dilution to determine the numbers of the cells. Typically, 10 µl of the diluted solutions were put onto the solidified LB medium plate, and incubated at 30 °C until the colonies became large enough to count. The experiments were replicated four times for each sample, and the numbers of the colonies on the LB plates were counted to calculate averages and standard deviations. This procedure

was performed at least twice. Successful removal of the biofilm cells from the wells was confirmed by staining the microtiter plates with CV, where the residual stain was negligible for all cases.

## Measurement of the total cellular concentrations of ATP

The cellular concentrations of ATP for the planktonic and biofilm cells for the WT and mutants were measured using a BacTiter-Glo Microbial Cell Viability Assay kit from Promega (Madison, WI, USA), and were expressed in relative light unit (RLU). The measurements for RLU were performed using a Lumitester C-100 from Kikkoman (Chiba, Japan). The planktonic and biofilm cells derived from the biofilm assay were used for the measurements, as described above. For the measurements of the planktonic cells, 100 µl of the supernatants were mixed with 100 µl of the BacTiter-Glo reagent. After incubated at ~5 min, RLUs were measured. For biofilm cells, the attached cells were finally diluted to 10-fold with the PBS buffer. Each 100 µl was used for the RLU measurements. The experiments were repeated four times for each sample, and the averages and standard deviations were calculated. The Student's $t$-test was performed to see difference between WT and the mutants.

## Measurement of cell surface LapA by dot-blot

Bacteria were grown statically in K10T-1 with or without 0.2 mM adenine for the indicated incubation period, or shaking at 250 rpm as indicated. Then cells were harvested and resuspended at a cell density of $OD_{600} = 0.25$ in PBS, and spotted in 5 µl aliquots onto nitrocellulose membrane filters. After drying, blots were blocked for 2 h with a 1% solution of Bovine serum albumin (BSA) in Tris-Buffered Saline (pH 7.4) containing 0.1 % Tween (TBST). Then blots were probed for the HA-tagged LapA protein with anti-HA antibody (Fisher Scientific, RB1438P0) at a concentration of 2 µg/ml in TBST 1% BSA. After 3 washes of 10 min each in TBST, blots were probed with a horse radish peroxidase-conjugated secondary antibody (PI-31460; Fisher Scientific, Hampton, NH, USA) at a concentration of 80 ng/ml in TBS, then rinsed, imaged and quantified as previously described (*Newell, Monds & O'Toole, 2009*).

## Scanning electron microscopy

Scanning electron micrographs were obtained for the biofilms formed inside the 96-well microtiter plate, followed by platinum ion sputtering (MSP-1S magnetron sputter, Vacuum Device Inc., Japan). A desktop scanning electron microscopy Phenom ProX (Phenom World, The Netherlands) was employed with the accelerating voltage of 10 kV. To obtain information on cell size, the electron micrographs were analyzed using Phenom ParticleMetric software (Phenom World, Eindhoven, The Netherlands). From each micrograph, single cells were automatically identified using the following parameters: minimum contrast, 0.50; merge shared borders, 0.25~0.35; conductance, 0.25~0.30; minimum detection size (%), 0.50. After manually removing the non-cell-derived small particles and the aggregated cells from the data collections, the histograms and parameters such as circumscribed circle diameter and circle equivalent diameter were automatically obtained. Five to eighteen SEM images were used to collect more than 1,000 of the

circumscribed circle diameters for each analysis, which depended on the numbers of the cells in each micrograph. Analyses were performed at least three times for each sample.

## Laser diffraction particle analysis

Size distributions of the planktonic and biofilm cells were measured using a laser diffraction particle analyzer LA-960 (Horiba, Kyoto, Japan). Particle size analysis was performed with the software equipped with the device. In this analysis, the software calculates the size of particles based on the Mie theory, which assumes the particles to be spherical (*Eshel et al., 2004*). Because the mutants did not grow well in K10T-1 medium, it was anticipated that the amounts of the cells were not enough for the measurements. We therefore used a prototype 200 μl volume microcell for the measurements, which is not commercially available currently. The biofilm formation assays were performed with 10 ml of K10T-1 in a standard 50 ml centrifugal tube made of polypropylene. After static incubation at 30 °C for 6 h, the planktonic cells were collected by centrifugation (7.4 k rpm for 2 min), and the cell-free supernatants were discarded so that the residual bacterial solutions were roughly 200 μl. The bacterial cells were resuspended using a micropipette, and then placed into the microcell using a disposable plastic syringe. For the measurement of biofilm cells, the 50 ml tubes were washed twice with the PBS buffer, and then 10 ml of the fresh buffers were added. The biofilm cells were dispersed using an ultrasound homogenizer VC-130 equipped with a 2 mm microtip (Sonics & Materials, Inc., Newtown, CT, USA). The amplitude was set to 20∼25%. The dispersed biofilm cells were collected by centrifugation and used for the measurements as described above. The measurements were repeated at least three times for each sample.

## RESULTS

### Mutants incapable of *de novo* purine biosynthesis show reduced biofilm formation, but are rescued by addition of adenine and hypoxanthine

Our previous transposon mutagenesis study indicated that the genes involved in the *de novo* purine nucleotide biosynthesis are required for normal biofilm formation by *P. fluorescens* Pf0-1 (*Newell et al., 2011*). To verify this, we constructed the clean deletion mutants for eight genes among total of eleven genes located in the pathway from PRPP to IMP, and examined their biofilm formation (Fig. 1A). Both PurN and PurT catalyze the same enzymatic step from 5-phosphoribosylglycinamide (GAR) to 5′-phosphoribosyl-*N*-formylglycinamide (FGAR). Since it was anticipated that the deletion of one of the genes would be compensated by the other, we did not make mutants for these genes. For the Δ*purB* mutant, we were unable to delete the gene despite repeated attempts. We therefore did not investigate these three genes in this study. As shown in Fig. 1A, biofilm formation by the mutants was reduced to less than half of that by WT, which confirms the transposon mutagenesis study.

To test whether these mutations were sufficient to explain the decrease in biofilm, we performed a complementation analysis, reintroducing each gene on a plasmid (Fig. 1B). Sufficient restoration of the biofilm formation was observed for complemented strains

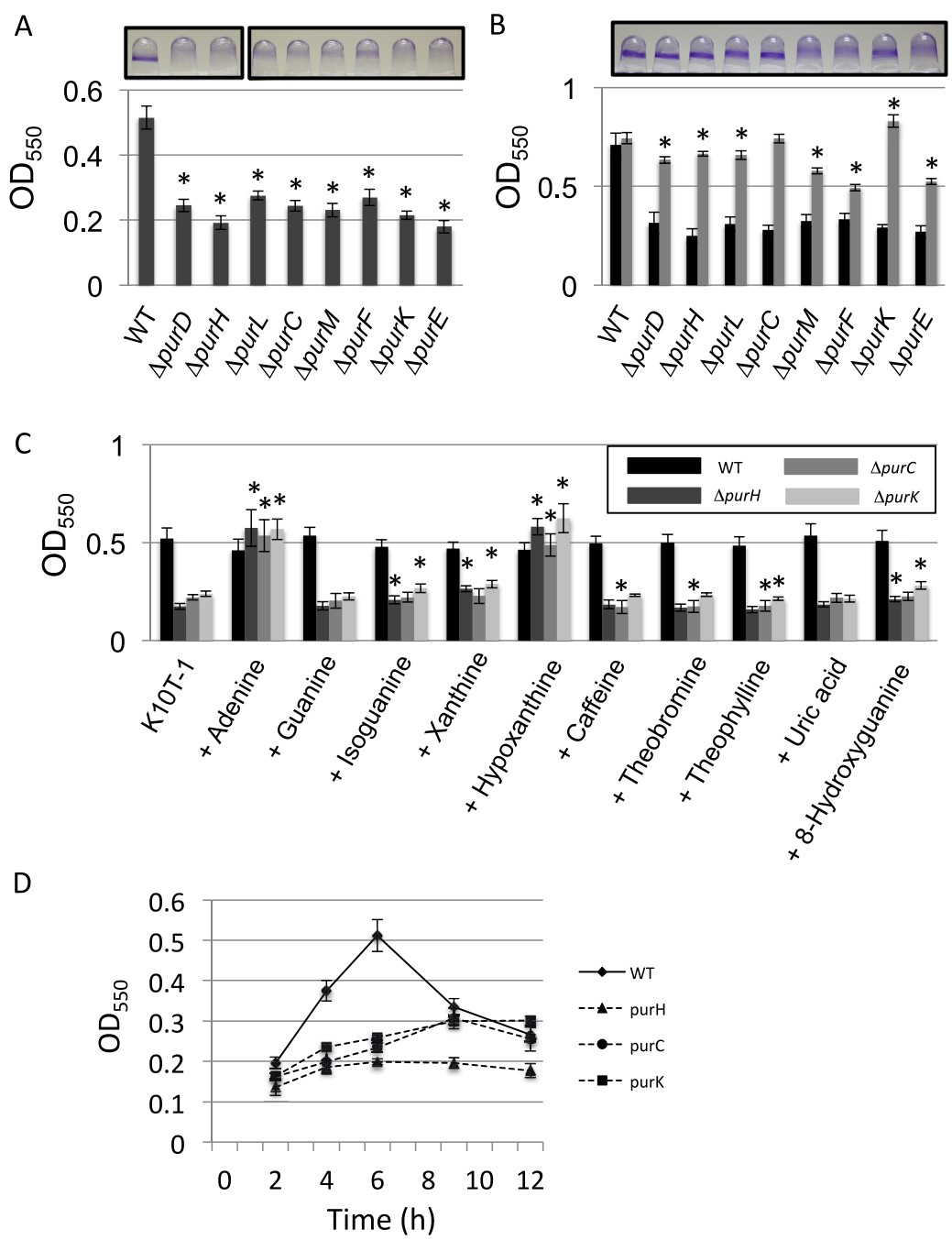

**Figure 1** **Effect of the mutation of the genes in the *de novo* purine nucleotide biosynthesis on biofilm formation.** (A) A quantitative biofilm assay comparing WT and the clean deletion mutants. Data are the mean absorbance of dissolved crystal violet stained biomass at 550 nm $\pm$ standard deviations (SD) ($n =$ 7). Representative images for the biofilms are shown above the graphs. The experiments were performed several times, and the representative one is shown. The asterisks ($*$) indicate significant differences in absorbance at 550 nm ($OD_{550}$) relative to that of WT ($P < 0.01$ in two-tailed Student's $t$-test assuming equal variance). (B) A quantitative biofilm assay to examine complementation 

**Figure 1 (…continued)**
of the genes for the clean deletion mutants (gray bars). The pMQ72 vector with each gene was introduced into the parent strain. The WT strain contained an empty pMQ72 vector. Arabinose was added to the medium at a concentration of 0.1% to induce the gene expression. Strains without the pMQ72 vector are shown in black bars as references. Data are the mean absorbance at 550 nm $\pm$ SD ($n = 7$). Representative images for the biofilms with the gene complementation are shown above the graphs. The asterisks ($*$) indicate significant differences in absorbance at 550 nm ($OD_{550}$) relative to that of WT ($P < 0.01$ in two-tailed Student's $t$-test assuming equal variance). The experiments were performed several times, and one of the results is indicated. (C) A quantitative biofilm assay for WT (black), $\Delta purH$ (dark gray), $\Delta purC$ (gray), and $\Delta purK$ (light gray) in the presence of various purine bases. Data are the mean absorbance at 550 nm $\pm$ SD ($n = 8$). Asterisks ($*$) show a statistically significant difference in absorbance relative to the medium without exogenous purines ($P < 0.01$ in two-tailed Student's $t$-test assuming equal variance). Each purine base was added to the medium at concentration of 0.2 mM. The experiments were done at least twice. (D) Time-dependent changes in biofilm formations by the WT and mutants strains in K10T-1. Data at each time point are the mean absorbance at 550 nm $\pm$ SD ($n = 7$). The experiments were done duplicate, one of which is shown.

of $\Delta purD$, $\Delta purH$, $\Delta purL$, $\Delta purC$, and $\Delta purK$. The recovery of biofilm formation was partial for complemented strains of $\Delta purM$, $\Delta purF$, and $\Delta purE$ (Fig. 1B). Instead, full recovery of biofilm formation for these three mutants was achieved by the addition of adenine (final conc. 1 mM) to the medium (Fig. S1).

We next examined the effect of adding purine bases to the growth medium on biofilm formation by WT and three mutants, $\Delta purH$, $\Delta purC$, and $\Delta purK$, was examined (Fig. 1C). Due to low solubility of xanthine in the medium, the concentration of the purines added to the K10T-1 medium were set to 0.2 mM in this experiment. In the salvage pathway, adenine, guanine, hypoxanthine and xanthine are converted to the corresponding nucleotides by adenine phosphoribosyltransferase and hypoxanthine-guanine phosphoribosyltransferase and xanthine phosphoribosyltransferase. In *E. coli* and *Salmonella*, it is known that addition of adenine, hypoxanthine, and guanine to the medium recovers the growth of the purine auxotroph mutants (*Neuhard & Nygaar, 1987*). This is because these bacteria possess adenosine deaminase and GMP reductase that convert adenosine and GMP to inosine and IMP, respectively, enabling the bacteria to synthesize both AMP and GMP from these purine bases.

The biofilm phenotypes shown in Fig. 1C indicated that only adenine and hypoxanthine were able to recover the same level of biofilm formation as that of WT. On the other hand, addition of guanine or uric acid did not affect biofilm formation. BLAST searches indicated that *P. fluorescens* Pf0-1 possesses a homologous gene (Pfl01_0671) to the adenosine deaminase of *E. coli* with sequence identity of 28%, but there was no hit for the GMP reductase, suggesting lack of the latter enzyme in this microorganism. Interestingly, xanthine and oxidized purines, isoguanine and 8-hydroxyguanine, slightly promoted biofilm formation by $\Delta purH$ and $\Delta purK$. Incorporation of xanthine into DNA and RNA was previously reported for *E. coli* mutants that cannot convert IMP to XMP or AMP (*Pang et al., 2012*). A similar mechanism may work for this case. In contrast, caffeine, theobromine and theophylline that are the methylated derivatives of xanthine were insensitive to the biofilm formation by WT, $\Delta purH$ and $\Delta purK$. Very minor inhibitory effect was found for $\Delta purC$ (Fig. 1C).

## Biofilm development of the mutants became slower than WT

As shown in Fig. 1D, the biofilm formation by WT reached maximum at around 6 h and then started to decline, which could be caused by dispersal of the cells due to nutrient deficiency (see *Newell et al., 2011a*). In contrast, the amounts of biofilms by the $\Delta purH$, $\Delta purC$, and $\Delta purK$ mutants reached maxima at around 9∼12 h (Fig. 1D), indicating that the biofilm development of the mutants were slower than WT.

## Mutations cause growth defect in the planktonic cells

One of the reasons for the reduced biofilm formation by the mutants could be a growth defect in the medium used in the assay, K10T-1. The supernatant for WT in the biofilm formation assay gradually became cloudy due to the growth of the cells. On the other hand, those for the mutants were clear and did not become turbid. To compare the growth of the cells during planktonic culture in this medium, changes in the absorbance at 590 nm ($OD_{590}$) were monitored for WT and the above three mutants ($\Delta purH$, $\Delta purC$, and $\Delta purK$) (Fig. 2A). It was apparent that the $OD_{590}$ for mutants did not change during 6 h, indicating that the mutants were not able to grow in K10T-1 medium. Addition of adenine into the medium rescued mutant growth, as the $OD_{590}$ started to rise sharply at around 4 h, as observed for WT (Fig. 2B). This result is consistent with the recovery of the biofilm formation in the presence of adenine, as shown in Fig. S1. It therefore seems likely that reduced biofilm formation by mutants in *de novo* purine biosynthesis pathway is due, in part, to the inability to grow in K10T-1 medium.

## Mutations lead to purine deprivation in both planktonic and biofilm cells

Since the mutants lack one of the genes in the biosynthesis pathway in the IMP synthesis, and demonstrate a growth defect in K10T-1 medium in the absence of purine supplementation, we predicted that the cellular concentration of purine nucleotides should be lower in the mutants than in the WT in K10T-1. Using the luciferase-based ATP assay we measured the relative concentrations of ATP in both planktonic and biofilm cells. In Fig. 2C, the black and grey bars indicate ATP contents for the planktonic and biofilm cells, respectively. The total ATP contents in the planktonic cells of the mutants were almost 20-times lower than that of the WT cells. Similarly, those in the surface-attached cells of the mutants were roughly 10-times lower than that of WT. These results indicate that the cellular concentrations of purines are likely to be reduced in the mutant cells.

## Purine biosynthesis mutations alter the proportion of attached verses planktonic cells in static culture

Although biofilm formation by the mutants was reduced to less than half of that by WT (Fig. 1A), it conversely suggests that certain numbers of mutant cells were attached to the surface as biofilms. To examine this more closely, the numbers of cells in the planktonic and attached populations were determined (Fig. 2D). It is known that LB medium contains substantial amounts of nucleic acids derived from yeast extract. Therefore, it is not surprising that the mutants grew well in LB medium, which was used to culture the inoculum for the biofilm assay (black bars in Fig. 2D).

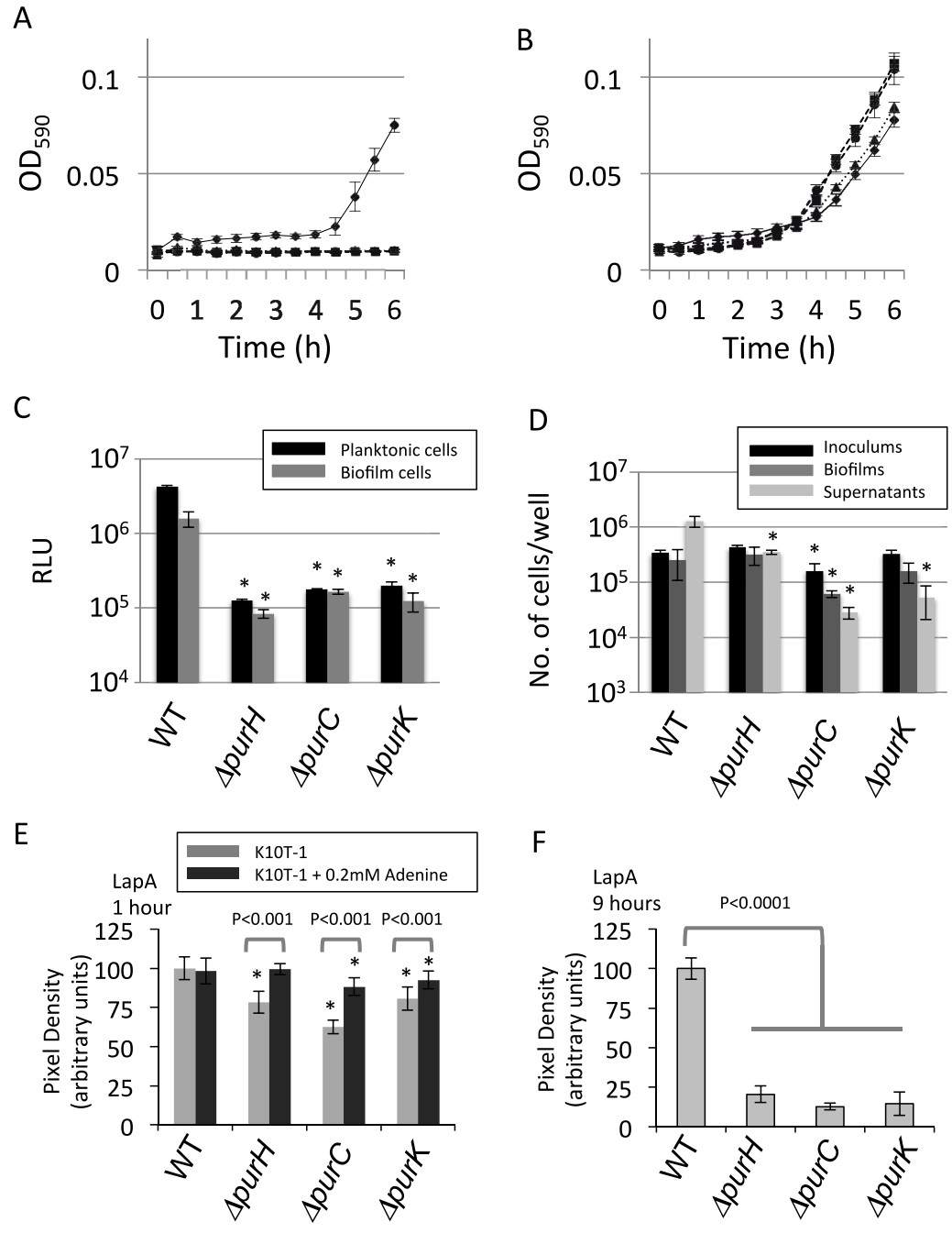

**Figure 2** **Analyses of the planktonic and biofilms cells and quantification of cell surface LapA.** (A) Time-dependent change in cell density for WT (♦) and the mutants ($\Delta purH$ (▲), $\Delta purC$ (●) and $\Delta purK$ (■)) in K10T-1 medium (absorbance at 590 nm; mean ± SD, $n = 8$). The experiments were performed in the same volume as the biofilm formation assay, and were done in at least twice. (B) Time-dependent change in cell density for WT (♦) and the mutants ($\Delta purH$ (▲), $\Delta purC$ (●) and $\Delta purK$ (■)) when 1 mM adenine was added to the medium (absorbance at 590 nm; mean ± SD, $n = 8$). The experiments were done duplicate. The asterisks (∗) show statistically significant difference in number of the cells compared to that of WT ($P < 0.01$ in two-tailed Student's $t$-test 
**Figure 2 (…continued)**
assuming equal variance). (C) Measurements of total amounts of ATP in planktonic and biofilm cells for WT and mutants. Data are the means RLU ± SD ($n = 4$). The asterisks (∗) show statistically significant difference in RLU relative to that of WT ($P < 0.01$ in two-tailed Student's $t$-test assuming equal variance). (D) Effect of the mutations on numbers of the cells in inoculums, biofilms and supernatants. The black, dark gray, and light gray bars indicate number of the cells in the inoculum ($t = 0$ h), biofilms ($t = 6$ h), and supernatants ($t = 6$ h), respectively, for WT and the three mutants ($\Delta purH$, $\Delta purC$ and $\Delta purC$); mean ± SD, $n = 4$. The experiments were done duplicate. Significant differences ($P < 0.01$) were determined by two-tailed Student's $t$-test assuming equal variance and indicated with asterisks (∗). (E) Cell surface LapA quantification at 1 h. Strains were grown statically in K10T-1 with or without 0.2 mM adenine for 1 h, then harvested and probed for LapA-HA by dot blotting. All mutants showed a significant reduction in LapA level compared to WT (*$P < 0.001$, Wilcoxon Sum Rank Test). Addition of adenine significantly increased LapA on all of the mutants, and in the case of $purH$ restored a WT level of LapA. (F) Cell surface LapA quantification at 9 h. Strains were grown for 9 h with aeration in K10T-1, then LapA measured as in (E). All mutants showed a significant reduction in LapA level compared to WT (*$P < 0.0001$, Wilcoxon Sum Rank Test).

Comparing dark and light grey bars in Fig. 2D provides information on the proportion of biofilm cells versus planktonic cells, respectively. The fraction of the total cells that were attached to the surface was larger for the $\Delta purC$ (~69%), $\Delta purK$ (~75%) and $\Delta purH$ (~48%) mutants, compared to that for WT (~16%). These data seem to indicate that the mutants exhibit a growth defect in planktonic culture in K10T-1, and are consistent with the growth curves shown in Fig. 2A. They are also consistent with the survival of cells under purine deprivation in the biofilm mode of growth.

Secondly, it should be noted that the cell numbers of the biofilms for the $\Delta purH$ and $\Delta purK$ mutants were almost the same as that of WT. Remember that the CV staining of $\Delta purH$ mutant biofilms was reduced to ~40% of the WT (Fig. 1A). One explanation for the reduction in biofilm biomass by the $\Delta purH$ mutant could be a reduction in cell size of the $\Delta purH$ mutant compared to that of WT. In contrast, the cell number from the biofilm of the $\Delta purC$ mutant was one-order of magnitude lower than the others. Therefore, reduction in cell number should also be considered as a contributor to reduced biofilm formation by the mutants. Altogether, these observations provided a hypothesis that the reduction in biofilms by the mutants is originated from the reduction(s) in cell sizes and/or cell numbers for the biofilm cells.

## Purine deprivation reduces amount of the surface adhesin, LapA in the planktonic cells

Biofilm formation by *P. fluorescens* Pf0-1 requires initial attachment via the cell surface adhesin, LapA (*Hinsa et al., 2003*; *Boyd et al., 2014*). The amount of this protein on the cell surface is controlled by internal signals via the binding of c-di-GMP to LapD (*Monds et al., 2007*; *Newell, Monds & O'Toole, 2009*). We next addressed whether the *pur* mutants were capable of attachment via LapA, despite showing reduced ATP levels which could influence cellular GTP and c-di-GMP concentrations. To test LapA levels during initial attachment, cells were incubated statically in K10T-1 medium for 1 h with or without adenine. Next, cells were harvested and probed for cell surface LapA by dot-blot. We found the LapA level was significantly reduced in the $\Delta purH$, $\Delta purC$ and $\Delta purK$ mutants compared to WT ($P < 0.001$, Wilcoxon Sum Rank Test), however the reduction

was not more than 40% (Fig. 2E). By comparison, deletion of *lapD* has been shown to reduce cell–surface LapA by >90% (*Newell, Monds & O'Toole, 2009*). Addition of adenine to the medium significantly increased LapA localization to the cell surface in all of the mutants ($P < 0.001$), but had no impact on LapA localization in the WT (Fig. 2E). Only in the case of the $\Delta purH$ mutant did adenine supplementation restore the LapA level to one indistinguishable from WT ($P > 0.5$), while the $\Delta purC$ and $\Delta purK$ mutants were partially restored to 88% and 92% of WT, respectively.

We hypothesized that further reduction in cell surface LapA might be observed for the mutants after prolonged incubation in the absence of adenine. To test this, cells were incubated in K10T-1 for 9 h under aerated conditions. These cells showed a greater reduction in cell surface LapA relative to WT than that seen during the initial attachment phase ($\Delta purH$ 20%, $\Delta purC$ 13%, and $\Delta purK$ 14% of WT level; Fig. 2F). Altogether, the LapA quantification results are consistent with the *pur* mutants attaching to surfaces via the LapA adhesin, albeit at a lower level than that of WT, and indicate that purine deprivation reduces the amount of LapA on the cell surface of planktonic cells.

## Cell size is altered by purine deprivation

From the cell counting experiments (Fig. 2D), the numbers of the surface-attached cells of the $\Delta purH$ and $\Delta purK$ mutants were not significantly different from that of WT, while the amounts of biofilms by the mutants were less than half of WT (Fig. 1A). To explain the reduction in the biofilm formation by the mutants, we hypothesized that the mutant cells in biofilms became smaller than WT. To test this, SEM images for WT and the three mutants cultured in K10T-1 were obtained (Fig. 3A). These data indicated that the amounts of the mutant cells in each view field of the SEM image were less than that of WT, which seems contradict to the results in the cell numbers (Fig. 2D). However, this is not unexpected because the mutant cells are seen at wider region in the wells of the microplate than WT whose biofilm cells are observed only at the air-liquid interface (Fig. 1A). To clarify the difference in the cell sizes between WT and the mutants, the SEM images were analyzed by measuring the circumscribed circle diameter of individual cells as an approximation for the length of the cell (described further in 'Materials & Methods'). We used the Wilcoxon Sum Rank test to compare the medians (Table 1). The medians for the circumscribed circle diameter for the $\Delta purH$, $\Delta purC$, and $\Delta purK$ mutants were 1.31, 1.35, and 1.43 μm, respectively, and became significantly smaller than WT (1.87 μm) by 25∼30%.

As the impaired biofilm formation by the mutants was restored by the supplementation of adenine (Fig. 1C), we expected that the cell sizes for the mutants would return to the equivalent size of WT. Figure 3B shows the electron micrographs and histograms obtained for the adenine-supplemented biofilm cells of WT and the mutants. Against our expectation, the WT cells showed a slightly smaller size (1.82 μm) compared to that (1.87 μm) in K10T-1 medium. Growth inhibition of *E. coli* by adenine supplementation has been known, which has been supposed to be attributed to inhibition of the *de novo* biosynthesis of pyrimidine nucleotides (*Hosono & Kuno, 1974*) or depletion of cellular concentrations of GTP (*Levine & Taylor, 1982*) and PRPP (*Shimosaka et al., 1984*). Similar

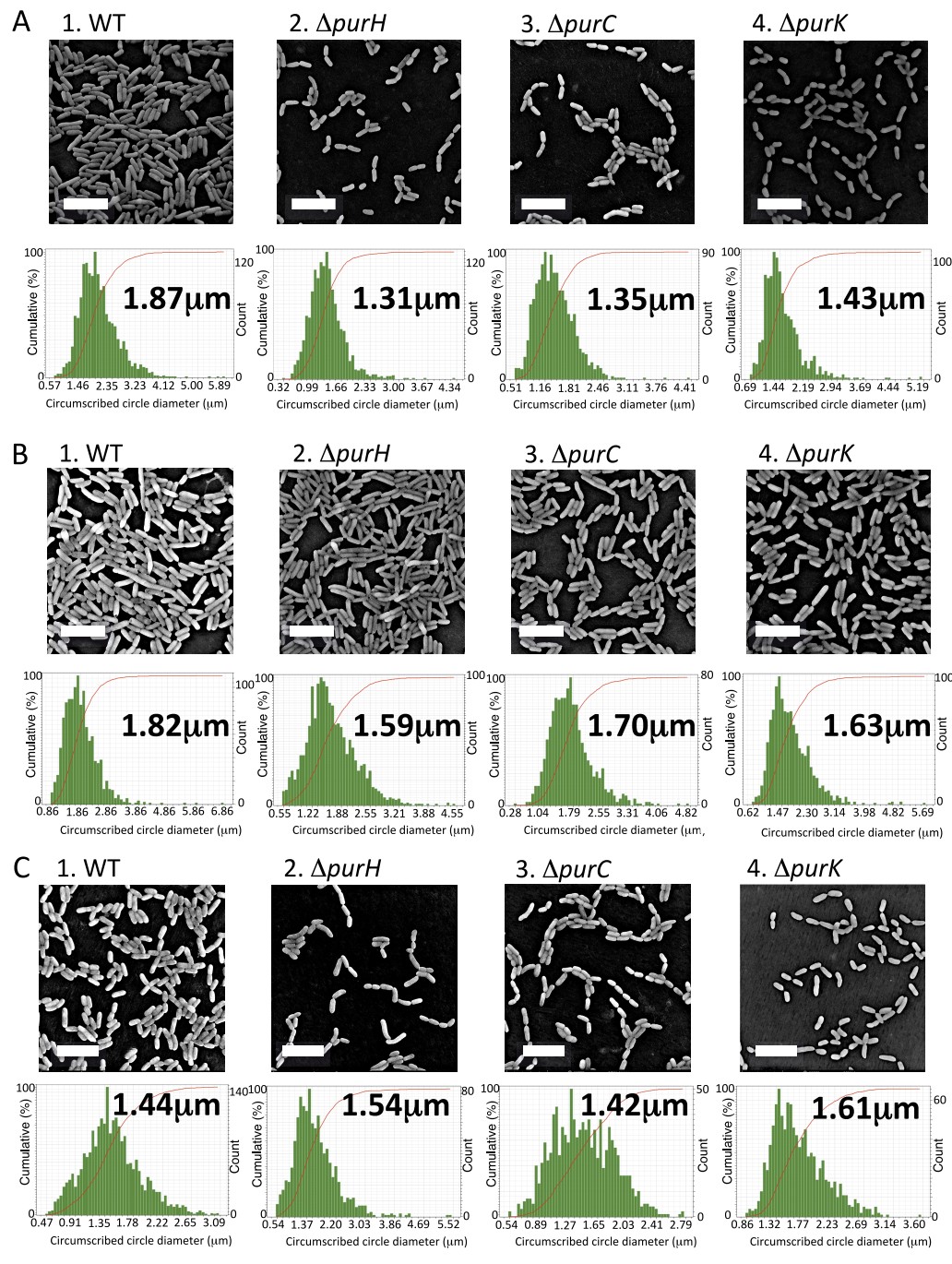

**Figure 3** **Effect of the mutations on sizes of the biofilm cells.** (A) Scanning electron micrographs showing the differences in cell size of the biofilm cells between WT and the mutants ($\Delta purH$, $\Delta purC$ and $\Delta purC$) in K10T-1 at 6 h. The white bars represent 5 µm. The histograms shown below the micrographs are the output data from the Phenom ParticleMetric software shown with minor modifications on the axes and labels for clarity. The medians of the circumscribed circle diameters are indicated on the each histogram. The analyses were performed three times for each strain, and one of them is shown in the figure. Five to fifteen SEM images were used to collect the circumscribed circle diameters of more than 1,000 cells for each analysis. (B) Scanning electron micrographs showing (continued on next page...)

**Figure 3 (…continued)**
changes in the cell sizes upon supplementation of 0.2 mM adenine into K10T-1 at 6 h. The white bars indicate 5 μm. The histograms labeled with the medians are shown below the micrographs. The analyses were performed three times for each strain, and one of them is represented. To collect the circumscribed circle diameters of over 1,000 cells, five to seven SEM images were used. (C) Scanning electron micrographs showing changes in the cell sizes of the biofilm cells when cultured at 9 h in K10T-1. The white bars indicate 5 μm. The histograms labeled with the medians are shown below the micrographs. The analyses were performed three times for each strain, and one of them is shown in the figure. Twelve to eighteen SEM images were used for the analyses.

**Table 1** Summary of the circumscribed circle diameters (μm) for WT and the mutants obtained by the SEM analyses.

|  |  | WT | Δ*purH* | Δ*purC* | Δ*purK* |
|---|---|---|---|---|---|
| K10T-1 at 6 h | Median | 1.87 | 1.31 | 1.35 | 1.43 |
|  | *P* (Wilcoxon) |  | (< 2.2E–16)[a] | (< 2.2E–16)[a] | (< 2.2E–16)[a] |
|  | Mean ± S.D. | 1.96 ± 0.59 | 1.35 ± 0.39 | 1.40 ± 0.42 | 1.53 ± 0.48 |
| +0.2 mM adenine at 6 h | Median | 1.82 | 1.59 | 1.70 | 1.63 |
|  | *P* (Wilcoxon) |  | (< 2.2E–16)[a] | (2.99E–10)[a] | (< 2.2E–16)[a] |
|  | *P* (Wilcoxon) | (0.002373)[b] | (< 2.2E–16)[b] | (< 2.2E–16)[b] | (< 2.2E–16)[b] |
|  | Mean ± S.D. | 1.89 ± 0.57 | 1.68 ± 0.56 | 1.76 ± 0.54 | 1.74 ± 0.56 |
| K10T-1 at 9 h | Median | 1.44 | 1.54 | 1.42 | 1.61 |
|  | *P* (Wilcoxon) |  | (2.3E–11)[a] | (0.1079)[a,*] | (2.20E–16)[a] |
|  | *P* (Wilcoxon) | (<2.2E–16)[b] | (<2.2E–16)[b] | (9.98E–05)[b] | (<2.2E–16)[b] |
|  | Mean ± S.D. | 1.48 ± 0.43 | 1.65 ± 0.60 | 1.45 ± 0.38 | 1.69 ± 0.40 |

**Notes.**
The parentheses indicate the *P* values obtained by the Wilcoxon Sum Rank test.
[a]*P* value when compared to the WT values in the same media.
[b]*P* value when compared to the same strain in K10T-1 medium at 6 h.
*The asterisks indicate no statistically significant difference in size relative to that of WT in the same medium ($P > 0.05$).

mechanisms may work for the WT strain of *P. fluorescens* Pf0-1, resulting in the observed slight shrinkage of the biofilm cells in the presence of adenine. The circumscribed circle diameters of the Δ*purH*, Δ*purC*, and Δ*purK* mutants when adenine was added to K10T-1 were 1.59, 1.70, and 1.63 μm, respectively. This represents a significant increase in mutant cell size due to the addition of adenine, ranging between 14 and 26%, suggesting that adenine supplementation can partially rescue the defect in cell size shown by the mutants (Table 1).

As stated above, biofilm formations by WT and the mutants showed time-dependent changes (Fig. 1D). At later than 6 h, the amounts of biofilms by the mutants gradually increased and came close to that of WT. These observations suggest that the biofilm cells of the WT become smaller while the mutant cells become larger due to the cell growth. To elucidate this point, we obtained the SEM images for the biofilm cells at 9 h, and the median sizes of the surface-attached cells were determined (Fig. 3C and Table 1).

WT cells became significantly smaller (1.44 μm) than that at 6 h (1.87 μm; $P < 0.01$), at longer incubation time. In contrast, the median sizes for the Δ*purH*, Δ*purC*, and Δ*purK* mutants were 1.54, 1.42, and 1.61 μm, respectively (Fig. 3C and Table 1), all

significantly larger than those at 6 h but still smaller than the original size (WT at 6 h, 1.87 µm; see $P$ values in Table 1). The increases in cell sizes (Fig. 3C) and amounts of biofilms (Fig. 1D) observed for the mutants may indicate gradual cell growth of the surface-attached cells despite purine limitation.

## Biofilm cells are smaller than planktonic cells

As revealed by the SEM analyses, the biofilm cells of the mutants possess smaller cell sizes than WT. The observation seems to be similar to that happens when bacteria are placed under condition like carbon starvation (Östling et al., 1993) or undergo what are typically called reductive divisions (Roszak & Colwell, 1987; Nyström, 2004). In the latter case, cell number increases without significant increase in biomass, which accompanies decrease in the cell size (Roszak & Colwell, 1987; Nyström, 2004). The reason for our observation may be ascribed to survival of the mutants on the surface under the purine limitation. This is an interesting hypothesis, but has not been confirmed yet. One way to test this is to know the sizes of the planktonic cells and compare those of the biofilm cells.

To get insights into the cell sizes for the planktonic cells, the size distributions for the planktonic cells were measured using a laser diffraction particle analyzer (Fig. 4). We compared the mode diameters that are the highest peak of the frequency distribution and represent the most commonly found cell (particle) sizes in the sample. To show distribution width for each analysis, D10, D50 (median), and D90 are shown in Table 2.

Shown in Fig. 4A are the frequency distributions for the planktonic cells of WT cultured in K10T-1 (solid line) and K10T-1 + adenine (dashed line). Note that the solid line almost overlapped the dashed line, indicating little change in the size of planktonic cells with or without adenine. The mode diameter for WT cells grown in both conditions was 1.86 µm. Biofilm cells were also analyzed after removal from the surface, which is shown in the dotted line in Fig. 4A. The mode diameter of WT cells growing in a biofilm was significantly smaller than that of the planktonic cells in K10T-1 (1.62 µm vs. 1.86 µm; $t$-test, $P < 10^{-10}$), suggesting that the change in life style from the planktonic to the surface-attached makes the cell size smaller for the WT cells (Table 2).

A reduction in the sizes of biofilm cells compared to planktonic cells was also observed for the mutants (Table 2). The mode diameter measurements of the mutant planktonic cells in K10T-1 medium ranged from 1.86 µm to 1.94 µm but were not significantly different from WT ($P < 0.05$; $t$-test). In contrast, the mode diameters of the mutant biofilm cells were significantly smaller than planktonic or WT biofilm cells. The mode diameters were 1.08 µm for the $\Delta purH$ and $\Delta purC$ mutants and 1.13 µm for the $\Delta purK$ mutant, which corresponds to 30~33% reduction in size compared to WT biofilm cells ($P < 0.001$, $t$-test) and a 42–44% reduction compared to the planktonic cells of each strain ($P < 0.002$, $t$-test). These results are consistent with the SEM analyses that indicated 25~30% of reductions in the circumscribed circle diameters for mutant biofilm cells compared to WT.

Supplementation of adenine had minimal effects on the size planktonic mutant cells, indicating that the planktonic cells of the mutants maintain the same cell size irrespective of purine levels in the growth medium (Table 2; difference from same strain without

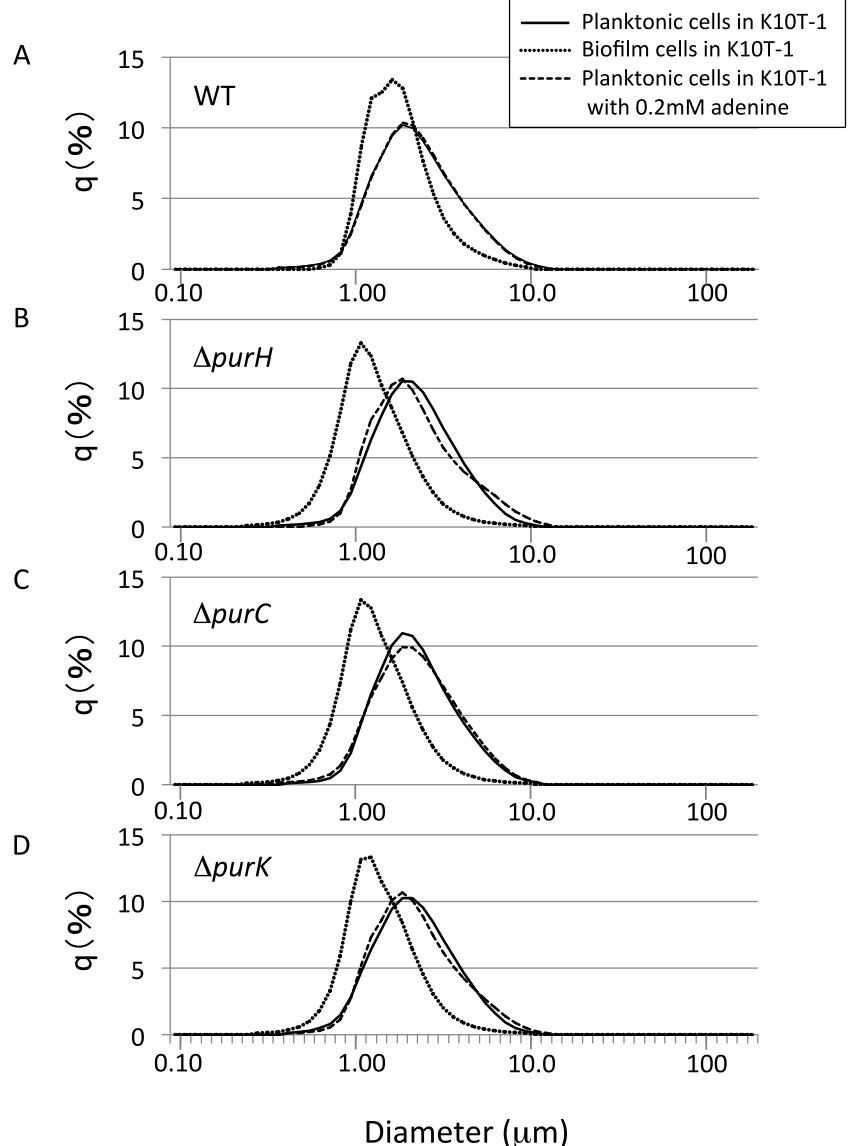

**Figure 4 The size distributions for the planktonic and biofilm cells of WT and mutants obtained using a laser diffraction particle analyzer.** (A–D) show the size distribution for WT, $\Delta purH$, $\Delta purC$, and $\Delta purK$, respectively. For each panel, the size distributions for planktonic and biofilm cells cultured in K10T-1 medium are shown in solid and dotted lines, respectively. The dashed lines indicate the size distributions for the planktonic cells cultured in the medium with 0.2 mM adenine. The measurements were performed three times for each, and one of the results are shown in the figure. See Table 2 for the statistics of the repeated measurements.

adenine not significant by $t$-test, $P > 0.05$). In summary, the laser diffraction particle analyzer data revealed significant differences between the size of planktonic and biofilm cells for the *pur* mutants and the WT. They also corroborate the observation by SEM that the surface-attached mutant cells have a smaller cell size than that of WT. Taken together with our analysis of CFU in the planktonic vs. attached phases of static cultures

**Table 2  Summary statistics for cell size of WT and mutants obtained by a laser diffraction particle analyzer.** Values represent mean ± standard deviation for three independent measurements.

| Conditions | Summary statistics | WT | ΔpurH | ΔpurC | ΔpurK |
|---|---|---|---|---|---|
| Planktonic in K10T-1 at 6 h | Mode size (μm) | $1.86 \pm 5 \times 10^{-4}$ | $1.94 \pm 0.14$ | $1.94 \pm 0.14$ | $1.94 \pm 0.15$ |
| | D10 (μm)[a] | $1.14 \pm 4 \times 10^{-3}$ | $1.16 \pm 2 \times 10^{-3}$ | $1.18 \pm 6 \times 10^{-3}$ | $1.12 \pm 9 \times 10^{-3}$ |
| | D50 (μm)[a] | $2.13 \pm 7 \times 10^{-3}$ | $2.14 \pm 2 \times 10^{-3}$ | $2.12 \pm 2 \times 10^{-2}$ | $2.12 \pm 6 \times 10^{-3}$ |
| | D90 (μm)[a] | $4.78 \pm 2 \times 10^{-2}$ | $4.40 \pm 4 \times 10^{-3}$ | $4.41 \pm 5 \times 10^{-3}$ | $4.38 \pm 4 \times 10^{-3}$ |
| | Mean size (μm) | $2.62 \pm 9 \times 10^{-3}$ | $2.52 \pm 1 \times 10^{-3}$ | $2.52 \pm 4 \times 10^{-3}$ | $2.49 \pm 9 \times 10^{-4}$ |
| Planktonic in K10T-1 + 0.2 mM adenine at 6 h | Mode size (μm) | $1.86 \pm 6 \times 10^{-4}$ | $1.85 \pm 6 \times 10^{-4}$ | $2.03 \pm 0.15$ | $1.85 \pm 2 \times 10^{-3}$ |
| | D10 (μm)[a] | $1.16 \pm 4 \times 10^{-4}$ | $1.15 \pm 2 \times 10^{-2}$ | $1.12 \pm 5 \times 10^{-3}$ | $1.14 \pm 1 \times 10^{-2}$ |
| | D50 (μm)[a] | $2.14 \pm 5 \times 10^{-3}$ | $2.04 \pm 1 \times 10^{-2}$ | $2.15 \pm 1 \times 10^{-2}$ | $2.07 \pm 2 \times 10^{-2}$ |
| | D90 (μm)[a] | $4.67 \pm 2 \times 10^{-2}$ | $4.94 \pm 4 \times 10^{-2}$ | $4.60 \pm 2 \times 10^{-2}$ | $4.71 \pm 5 \times 10^{-2}$ |
| | Mean size (μm) | $2.59 \pm 9 \times 10^{-3}$ | $2.62 \pm 2 \times 10^{-2}$ | $2.57 \pm 1 \times 10^{-3}$ | $2.56 \pm 2 \times 10^{-2}$ |
| Biofilm in K10T-1 at 6 h | Mode size (μm) | $1.62 \pm 6 \times 10^{-4*}$ | $1.08 \pm 6 \times 10^{-4*,**}$ | $1.08 \pm 6 \times 10^{-5*,**}$ | $1.13 \pm 8 \times 10^{-2*,**}$ |
| | D10 (μm)[a] | $1.08 \pm 3 \times 10^{-3}$ | $0.72 \pm 1 \times 10^{-3}$ | $0.76 \pm 5 \times 10^{-4}$ | $0.77 \pm 3 \times 10^{-2}$ |
| | D50 (μm)[a] | $1.69 \pm 6 \times 10^{-3}$ | $1.20 \pm 1 \times 10^{-3}$ | $1.25 \pm 4 \times 10^{-4}$ | $1.32 \pm 3 \times 10^{-2}$ |
| | D90 (μm)[a] | $3.15 \pm 2 \times 10^{-2}$ | $2.37 \pm 1 \times 10^{-3}$ | $2.43 \pm 3 \times 10^{-4}$ | $2.49 \pm 3 \times 10^{-2}$ |
| | Mean size (μm) | $1.98 \pm 9 \times 10^{-3}$ | $1.44 \pm 1 \times 10^{-3}$ | $1.49 \pm 3 \times 10^{-4}$ | $1.52 \pm 3 \times 10^{-2}$ |

Notes.

[a]The D50 is the median that is defined as the diameter where half of the population lies below this value. In the same way, 10 percent of the distribution lies below the D10, and 90 percent of the population lies below the D90. Standard deviations for D10 and D90 values were all below 0.05.

*Significantly smaller than planktonic cells of the same strain grown in K10T-1 by Student's $t$-test, $P < 0.002$.

**Significantly smaller than WT biofilm cells by Student's $t$-test, $P < 0.001$.

(Fig. 2D), these data suggest major differences in the physiology of attached cells under purine limitation compared to their free-floating counterparts.

## DISCUSSION

In this study, we sought to determine the basis for the defects in the biofilm formation by the purine auxotrophic mutants of *P. fluorescens* Pf0-1, in which one of the genes involved in the *de novo* purine biosynthesis pathway to IMP was disrupted. We found that the attached biomass in the mutant biofilms was less than half of that in WT biofilms in K10T-1 medium (Fig. 1A). As this biosynthesis pathway is essential for most of bacteria, many studies have shown that the disruption of the genes in this biosynthesis pathway also impact virulence, biofilm formation and symbiosis. Our results confirm that the purine auxotrophic mutants of *P. fluorescens* Pf0-1 show a reduced biofilm phenotype and provide details as to the mechanisms involved.

Synthesizing our results, we propose that the biofilm phenotype of the *pur* mutants is influenced by several factors. First, the mutations resulted in modest but significant reductions in cell surface LapA (Fig. 2E), which likely reduces the number of cells that initially attach to the surface. This is consistent with the lower density of attached cells observed by SEM (Fig. 3). Second, as the exogenous purine supply is exhausted during
static growth in K10T-1 the proliferation of the mutants is likely slowed due to purine limitation; this judgment is based on planktonic growth and biofilm cell counts (Figs. 2A and 2D). The growth defect observed in the absence of purine supplementation is one major contributing factor in the decrease in biofilm formation.

A third factor influencing the biofilm phenotype of the *pur* mutants is changes in cell size. Under purine limitation, attached cells became reduced in size, about 25–30 % smaller than WT by SEM (Fig. 3). This is consistent with the results from the laser diffraction particle analyses which also indicated a 30–33% size reduction for the mutants (Fig. 4 and Table 2), while the two methods utilize different theories to obtain the cell sizes. The attached mutant cells were significantly smaller than attached WT cells at 6 h when the greatest difference in attached biomass is observed by the CV assay (Fig. 1D). The amount of attached biomass and biofilm cell sizes for the WT and mutants converge at 9 h, likely due to the gradual elongation of mutant cells and potentially shortening of WT cells, which we predict experience some nutrient limitation by this time (Figs. 1D and 3).

One interesting finding of this study is the difference between attached and planktonic mutant cells. These differences are evident in the change in cell size upon purine limitation (Fig. 4), as well as in the larger number of viable cells recovered from the biofilm compared to the planktonic population for Δ*purC* and Δ*purK* (Fig. 2D). Additionally, the size of biofilm cells increased in response to adenine supplementation, while the planktonic cells did not (Fig. 3 vs. Fig. 4). Combined with the modest increase in mutant biofilm cell size between 6 and 9 h, and ATP measurements (Fig. 2C), these data argue that purine-limited biofilm cells are metabolically active.

Although the result in Fig. 2D indicated that the number of the biofilm cells was decreased one-order of magnitude for the Δ*purC* mutant, the ATP assay revealed that the total amount of ATP for this mutant was similar to those of the Δ*purH* and Δ*purK* mutants (Fig. 2C). These results may indicate that there are some problems in recovery process for the Δ*purC* mutant after plating on a LB medium plate in the cell counting experiments. In other words, the actual number for the biofilm cells for the Δ*purC* mutant may be higher than the value indicated by plate counts (Fig. 2D) and thus closer to those of the other mutants.

The laser diffraction study further revealed that the biofilm cells of WT are smaller than those in the planktonic phase of the same medium (Fig. 4 and Table 2). A similar observation was reported for *Staphylococcus aureus*, in which differences in total cellular proteins and respiratory activity between surface-attached and planktonic cells were observed (*Williams et al., 1999*). Therefore, our observation suggests that some metabolic differences exist between the two states. In addition, the cells sizes of WT biofilms in the presence of adenine are smaller than those in its absence (Fig. 3B and Table 1). For the purine auxotrophic mutants, the salvage pathway is the sole way to synthesize the purine nucleotides. However, for WT, the excess purine base in the medium not only suppresses the *de novo* purine biosynthesis pathway (*Houlberg & Jensen, 1983*) but also may influence other metabolic pathways, leading to the observed shrinkage of the surface-attached cells.

Further study using metabolomic and/or gene expression analyses is desired to identify the mechanism behind these changes.

In contrast to the surface-attached cells, the planktonic cells of the mutants remain the same cell size as WT irrespective of adenine supplementation (Fig. 4 and Table 2). As shown in Fig. 2D, the numbers of the planktonic cells for the mutants were less than the inoculums, indicating cell death occurs for some planktonic cells that fail to adapt to the purine deficiency. The behavior of the planktonic cells of the mutants is different from those of *Vibrio* sp. and *P. putida* KT2442, in which the cell size reduction occurred for the planktonic cells soon after they were placed under carbon limitation (*Amy & Morita, 1983*; *Östling et al., 1993*; *Givskov et al., 1994*). This is an interesting observation as it indicates the surface-attached forms of the mutants are more suitable for survival rather than the planktonic ones under the purine shortage.

## CONCLUSIONS

This study examined purine auxotrophic mutants of *P. fluorescens* Pf0-1 to elucidate the basis of the defects they display in biofilm formation. We found significantly more viable mutant cells attached to the surface than in the planktonic phase, indicating that the surface-attached mode of growth is suitable for survival of the mutants under the purine shortage. ATP measurements and the observed increases in cell size between 6 and 9 h also suggest that mutant biofilm cells are metabolically active. Using SEM and a laser diffraction particle analyzer, we demonstrated that the surface-attached mutant cells have smaller sizes than WT and that the surface-attached WT cells were smaller than the planktonic ones. The latter observation indicates that some modulation of cell size is a natural response of *P. fluorescens* to the biofilm environment. The reduction in the cell numbers and sizes in mutant biofilms could be main factors to explain the reduced biofilm formation by the mutants. The data presented here provides a new view on the relationship between purine deficiency and biofilm formation.

**Abbreviations**

| | |
|---|---|
| **CV** | Crystal violet |
| **SEM** | Scanning electron microscopy |

## ACKNOWLEDGEMENTS

We thank Shingo Shimoyama, Shoko Wada, and Atsumi Ozaki (Jasco International Co., Ltd., Japan) for their assistance in obtaining and analyzing the scanning electron micrographs. We are grateful to Kyoko Mitsunari and Kazuhiro Yoshida (Horiba Ltd., Japan) for the measurements using a laser diffraction particle analyzer. We also thank Dr. George O'Toole and two anonymous reviewers for critical reading of the manuscript.

### Funding

This research was supported by JSPS KAKENHI 23651226 (to SY) and a grant for Scholarly and Creative Activity from the Provost of SUNY Oswego (to PDN). The funders had no role in study design, data collection and analysis, decision to publish, or preparation of the manuscript.

### Grant Disclosures

The following grant information was disclosed by the authors:
JSPS KAKENHI: 23651226.
Provost of SUNY Oswego.

### Competing Interests

The authors declare there are no competing interests.

### Author Contributions

- Shiro Yoshioka conceived and designed the experiments, performed the experiments, analyzed the data, contributed reagents/materials/analysis tools, wrote the paper, prepared figures and/or tables, reviewed drafts of the paper.
- Peter D. Newell performed the experiments, analyzed the data, contributed reagents/materials/analysis tools, wrote the paper, prepared figures and/or tables, reviewed drafts of the paper.

### Data Availability

Figshare
http://dx.doi.org/10.6084/m9.figshare.1606256
http://dx.doi.org/10.6084/m9.figshare.1606257
http://dx.doi.org/10.6084/m9.figshare.1606258
http://dx.doi.org/10.6084/m9.figshare.1606259
http://dx.doi.org/10.6084/m9.figshare.1606260
http://dx.doi.org/10.6084/m9.figshare.1606261
http://dx.doi.org/10.6084/m9.figshare.1606262
http://dx.doi.org/10.6084/m9.figshare.1606263
http://dx.doi.org/10.6084/m9.figshare.1606264
http://dx.doi.org/10.6084/m9.figshare.1606265
http://dx.doi.org/10.6084/m9.figshare.1606266
http://dx.doi.org/10.6084/m9.figshare.1606294

### Supplemental Information

Supplemental information for this article can be found online at http://dx.doi.org/10.7717/peerj.1543#supplemental-information.

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
