# Peer review of "Disruption of de novo purine biosynthesis in Pseudomonas fluorescens Pf0-1 leads to reduced biofilm formation and a reduction in cell size of surface-attached but not planktonic cells"

_PeerJ, doi:10.7717/peerj.1543_

## Round 0.1 · original submission · Major Revisions

Your manuscript has been reviewed by two independent reviewers. Based on their comments, it seems that more experimental data are needed and the manuscript need to be edited extensively. We encourage to submit the revised manuscript.

Reviewer 1 ·

Basic reporting

OK

Experimental design

OK

Validity of the findings

No Comments

Additional comments

The manuscript by Yoshioka and Newell describes the impact of disruption of de novo purine biosynthesis pathway on biofilm formation. Using different mutants for genes involved in this pathway, they have demonstrated that the reduction in cell size of mutant Pf0-1 is the cause of reduced biofilm formation. The authors also claim that defect in de novo purine biosynthesis modulates the ratio of biofilm vs. planktonic cells. The manuscript is clearly written. I consider acceptance of the manuscript if the following points are addressed.

Major comments:

1) The relationship between disruption of de novo purine biosynthesis and cell adhesion required for biofilm formation was not properly established. For example, a measurement of surface expression or total amount of LapA or LapD in WT and mutants would have strengthened the data.
2) Level of purines in in each mutant has not been shown in this study. In one approach, a measurement of c-di-GMP can be done to show the defects in de novo purine biosynthesis in each mutant.
3) Results from figure 2 seem contradictory to each other. Figure 2A shows that the number of cells in biofilms is similar in WT vs. mutants however the images in figure 2B shows that the number of cells in mutants is significantly less than that of WT. A more careful analysis of cell number in biofilms for WT and mutant should be performed.

Minor comments:

1) For figure 1C, it was not explained why the addition of Guanine did not restore biofilm formation by the mutants.
2) The authors did not explain why further experiments were not performed to demonstrate full recovery of biofilm formation for ΔpurM, ΔpurF and ΔpurE in the complementation experiments.
3) Although the cell size of mutants in biofilms is smaller than WT, the number of cells for mutants is also less significantly compared to WT. Why only the cell size but not cell number, has been emphasized as important effect of disruption of de novo purine biosynthesis.
4) It was not properly explained why the cell size of WT was reduced when exogenous Adenine was supplemented as shown in figure 2.
5) In all the figure legends, it should be mentioned how many times each experiment was repeated.

Reviewer 2 ·

Basic reporting

This paper describes the importance of purine nucleotide biosynthesis in biofilm formation and extends this to P. fluorescens (Pf0-1). The authors have shown that deletion of 8 genes involved in this pathway individually leads to reductions in biofilm formation, and this can be rescued by purine addition or gene complementation. The authors claim that more viable mutant cells were recovered from the surface-attached population than in the biofilm/planktonic layer and that surface-attached mutant cells were 25~30% shorter in length than WT.
The importance of purine biosynthesis in bacterial biofilm formation is a known fact and I don’t see any novelty in extending this aspect to P. fluorescens (Pf0-1). Besides, other than basic reporting of the existence of greater percentage cells as surface attached and of smaller size, in-depth analysis into the mechanisms involved in these phenotypes is lacking.

Experimental design

There are certain drawbacks in the experimental designs as follows:
1. Crystal violet staining is a good indicator of biomass formation at the biofilm, but not a good indicator of cell viability. Staining using CFDA/PI will be a better measure of cell viability. This is critical when we attempt to determine the percentage of viable cells in WT v/s mutants and in planktonic v/s surface attached layers.
2. Most of the plots used in this study lack statistical tests and hence it is difficult to determine the significance of the data.
3. The mutant strains may simply be slow growing hence they show smaller size and lesser biofilm formation in 6hrs (time point used in this study). Hence a longer time point (around 24 hrs) will be useful in verifying/negating this aspect.
4. The concentration of adenine supplementation is not constant throughout the study (e.g. In Fig 1E conc. used is 1mM while in Fig 2C adenine used is 0.2mM.) Either concentration should be kept constant, or reasons for different dosage should be justified. Besides, the physiological concentration of adenine in WT cells will be worth mentioning in the text.

Validity of the findings

My major concerns regarding the findings are as follows:
1) The authors have mentioned having created “clean mutants” for this study. But there is no evidence presented to support the extent of depletion in purine biosynthesis in these mutants.
2) Fig 2A representing the reduction in biofilm formation by the mutants lacks statistical significance test and it appears that there is no significant difference in Inoculum v/s Biofilm or Supernatant cells. Fig 2B and Fig 2C representing SEM analysis of biofilm size should be from multiple view fields. However, it is unclear how many view fields were considered for these analysis.
3) The reduction of size shown by particle distribution analysis in Fig 3 doesn’t appear to be significant. Hence it is difficult to conclude that biofilm cells are of smaller size in mutants.

Additional comments

Besides, taking care of the above mentioned points, the authors needs to add certain concrete data to indicate the metabolic difference in the mutants v/s WT cells in purine deficient and purine supplemented conditions. The paper will gain the necessary weightage for publication when the authors experimentally establish the mechanism behind biofilm formation defect.

---

## Round 0.2 · accepted · Accept

Your revised manuscript has been reviewed by two independent experts and both have expressed their satisfactions on the revised version. I am happy to inform you that your revised manuscript has been now accepted now. Please ensure that your manuscript is free from any grammatical and spelling errors at this stage.

Reviewer 1 ·

Basic reporting

No Comments

Experimental design

No Comments

Validity of the findings

No Comments

Additional comments

The revised manuscript has been significantly improved. I consider acceptance of the manuscript for publication in PeerJ.

Reviewer 2 ·

Basic reporting

In the revised version, the Authors have addressed all major concerns raised by the reviewers.

Experimental design

No Comments

Validity of the findings

No Comments